# Cohort profile: Born in Wales—a birth cohort with maternity, parental and child data linkage for life course research in Wales, UK

Hope Jones [1], Mike J Seaborne [1], Natasha L Kennedy,[1,2]
Michaela James [1], Sam Dredge,[1] Amrita Bandyopadhyay [1],
Adele Battaglia [3], Sarah Davies,[4,5] Sinead Brophy [1,2,6]

¹National Centre for Population Health and Wellbeing Research, Swansea University Medical School, Swansea, UK
²Swansea University Medical School, Administrative Data Research Wales, Swansea, UK
³Patient and Public Involvement Representative, Swansea, UK
⁴Betsi Cadwaladr University Health Board, Bangor, UK
⁵Health and Care Research Wales, Cardiff, UK
⁶Health Data Research UK, London, UK

**Correspondence to**
Hope Jones;
H.E.Jones@Swansea.ac.uk

## ABSTRACT

**Purpose** Using Wales's national dataset for maternity and births as a core dataset, we have linked related datasets to create a more complete and comprehensive entire country birth cohort. Data of anonymised identified persons are linked on the individual level to data from health, social care and education data within the Secure Anonymised Information Linkage (SAIL) Databank. Each individual is assigned an encrypted Anonymised Linking Field; this field is used to link anonymised individuals across datasets. We present the descriptive data available in the core dataset, and the future expansion plans for the database beyond its initial development stage.

**Participants** Descriptive information from 2011 to 2023 has been gathered from the National Community Child Health Database (NCCHD) in SAIL. This comprehensive dataset comprises over 400 000 child electronic records. Additionally, survey responses about health and well-being from a cross-section of the population including 2500 parents and 30 000 primary school children have been collected for enriched personal responses and linkage to the data spine.

**Findings to date** The electronic cohort comprises all children born in Wales since 2011, with follow-up conducted until they finish primary school at age 11. The child cohort is 51%: 49% female: male, and 7.8% are from ethnic minority backgrounds. When considering age distribution, 26.8% of children are under the age of 5, while 63.2% fall within the age range of 5–11.

**Future plans** Born in Wales will expand by 30 000 new births annually in Wales (in NCCHD), while including follow-up data of children and parents already in the database. Supplementary datasets complement the existing linkage, including primary care, hospital data, educational attainment and social care. Future research includes exploring the long-term implications of COVID-19 on child health and development, and examining the impact of parental work environment on child health and development.

## STRENGTHS AND LIMITATIONS OF THIS STUDY

⇒ Born in Wales has established a comprehensive, Wales-wide population-based database which consolidates clinical data from maternity, neonatal, child health and education records.

⇒ This national-scale database is enriched by quantitative and qualitative results from surveys conducted by Born in Wales, providing rich insights into details that cannot be obtained through routinely collected data.

⇒ The existence of this database enables further data linkage, facilitating life course research on the health and well-being of the Wales population.

⇒ Missing data or errors in routine and administrative data may be a constraint.

⇒ A potential restriction of Born in Wales is the loss of data pertaining to individuals who relocate outside of Wales during pregnancy or after the child's birth.

This longitudinal approach encompasses the stages of pregnancy planning, pregnancy, childhood, adolescence and adulthood.[2] Birth cohorts such as Born in Bradford (BiB),[3] the eLIXIR Partnership,[4] and Born in Scotland[5] have contributed substantial evidence in the field of life course research, particularly focusing on pregnancy and early childhood.

Traditional cohorts are formed by recruiting participants within a specified timeframe. However, these cohorts can very quickly become outdated due to fluctuations in population demographics, lifestyle factors and environmental variations. Follow-up of such cohorts becomes challenging, as sample attrition, where participants withdraw, leave or do not complete the study, introduces methodological biases that can impact research validity and findings.[6] Furthermore, cohort recruitment may suffer from volunteer bias or self-selection bias, potentially limiting

## INTRODUCTION

The recognition of health promotion early in life is acknowledged as a strategy to enhance health and well-being across the life course.[1]

the representativeness of the cohort.[7] Population-based cohorts eliminate these biases and provide broader information about a wider scale of outcomes that may not be feasible in other cohort studies.[8]

Previous birth cohorts have faced challenges related to attrition and inclusion bias. Routine electronic health records, which encompass data on all women receiving antenatal care in Wales and their infants, offer a solution to reduce these complications.[9] While national birth cohorts have been well established in other countries,[10 11] the linkage of maternal and child data has not been extensively utilised in the UK. In Scotland, population data linkage has yielded findings that have informed UK clinical guidelines regarding maternal and neonatal outcomes.[12] Numerous successful linkages, such as maternity data with national birth registration datasets, birth registration with Hospital Episode Statistics, and utilisation of primary care pregnancy data in the UK,[13–17] have demonstrated the utility of this approach.

Born in Wales facilitates a platform for linking research datasets with authorised levels of patient anonymity and assured data security. Additional advantages include the capability to consolidate various datasets from other organisations, including healthcare, social care and education. Ethical guidelines must be adhered to when working with these datasets to ensure accurate data linkage and participant anonymity.

The Secure Anonymised Information Linkage (SAIL) Databank[18 19] plays a vital role in integrating and harmonising these datasets. This document outlines the safeguards implemented to protect user rights throughout the development and application of Born in Wales, as well as providing the demographics of the Born in Wales cohort. These procedures and mechanisms are established based on the successes observed in other databases and cohorts.[3–5]

Data-linkage facilitated by Born in Wales provides a unique data repository for investigating significant public health questions with far-reaching implications. The capacity to run these linkages permits a wide range of longitudinal health, social and education data to be collected along with the ability to enhance life course data analysis. Born in Wales operates as a longitudinal database, commencing in pregnancy, and incorporating routinely collected clinical data from maternity, neonatal, social care and education records, not solely relying on participant recruitment. Moreover, Born in Wales has the added element of enrichment by survey responses as well as the population-level data outcomes.

The core dataset for Born in Wales is the National Community Child Health Database (NCCHD) which provides child birth information, such as birth weight, gestation time, birth order, as well as information regarding the mother (eg, smoker, type of maternity care). This database is updated quarterly. With over 30 000 annual new births in Wales, Born in Wales has the potential to evolve into one of the most comprehensive datasets on parental and child health.

This document outlines how we have used the NCCHD as a core dataset to build a more comprehensive and complete maternity and birth cohort in Wales which can be linked and used for future longitudinal life-course research analyses.

## COHORT DESCRIPTION
### Data sources
Born in Wales establishes linkages between electronic data pertaining to mothers, babies and partners, including self-identified biological fathers or mothers' partners, for children born in Wales, encompassing approximately 30 000 annual births. The core dataset is NCCHD, which can be linked to various other records. The linked health records encompass primary care data (from Wales Longitudinal General Practice), secondary care (from hospital admissions, emergency care, inpatient from Patient Episode Database for Wales (PEDW), and outpatient from Outpatient Database for Wales), maternal indicators (midwife data), and public health records (vaccination uptake, hearing checks, health visitor assessment, breastfeeding initiation and duration, and COVID-19 vaccination/testing). Additionally, data pertaining to education, Census 2011, police/domestic violence, substance abuse, social care (looked after children, child protection register, children in receipt of care, and family court) have been successfully linked. Moreover, connections are being established to Census 2021 and the National Neonatal Audit database. Ultrasound scans conducted across Wales have been coded to identify follow-up markers predictive of future cognitive development and school readiness; these have also been prepared for integration into the Born in Wales data spine.

Maternity and neonatal data were acquired from the SAIL Databank, extracting from multiple datasets including the NCCHD, Maternal Indicators Dataset (MIDS) and PEDW. Refer to table 1 for further details regarding the datasets already linked or planned to be linked in Born in Wales.

In addition to the electronic birth cohort, the data spine is enriched with repeated surveys administered to both mothers and identified biological fathers or female/male partners during pregnancy, and children in primary school. These surveys capture self-reported health data pertaining to stress, mental health, occupation, ethnicity and open-ended questions addressing strategies to enhance health and well-being for families.[20] Since 2016, 30 000 primary school children have completed the Health & Attainment of Pupils in Primary EducatioN (HAPPEN) Survey. Primary schools are recruited via email and local authority referral. The survey is completed online, in school, at a time that suits the school. The survey has been developed alongside the new school curriculum for Wales priorities for health and well-being. Since 2020, 2500 parents have been recruited via social media to complete the Born in Wales surveys. This survey is also completed

**Table 1** Datasets linked or to be linked in Born in Wales

| Datasets | Example of data fields |
|---|---|
| **Electronic data** | |
| National Community Child Health Database (NCCHD) | Breastfeeding duration, birth weight, gestation, blood test results, maternal smoking, hearing/vision tests, health visitor assessment of family resilience, domestic violence, speech and language skills, flying start services. |
| Annual District Birth Extract (ADBE) | Register of all births and stillbirths (cause of death) in Wales, multiple births, Lower layer Super Output Area (LSOA) of mother, place of birth, no father on registration, marriage status of parents, 10% fathers' socioeconomic status. |
| Congenital Anomaly Register and Information Service (CARS) | Congenital anomaly (ICD10 code), gender, LSOA, information about mother including substance abuse, migrant status, diabetes, epilepsy. |
| Education Wales (EDUW) | Attainment (foundation phase, key stages 1 and 2), free school meal, attendance, special educational needs, education other than school, ethnicity, age, gender, local authority. |
| Patient Episode Dataset for Wales (PEDW) | Admission to hospital diagnosis and operation codes |
| Outpatient Dataset for Wales (OPDW) | Attendance information for all National Health Service (NHS) Wales hospital outpatient appointments |
| Maternal Indicators Dataset (MIDS) | Maternal data at initial assessment including age, smoking, weight, mental health condition, previous births, Labour and birth data including Apgar Score, birth outcome (c-section, assisted delivery, presentation for example, transverse), birth weight, intention to breast feed. |
| ONS 2011 Census Wales (CENW) (2021 Census to be linked) | Ethnicity, household composition, occupation, qualifications, place of work, long term health problem or disability, house data (number of rooms), gender, LSOA. |
| Children Receiving Care and Support Census (CRCS) | On child protection register, abuse type, disability, ethnicity, autism, parental capacity, dental health, mental health |
| Substance Misuse Dataset (SMDS) | Gender, age, substance abuse problem, parental responsibility, accommodation, mental health. |
| Children and Family Court Advisory and Support Service (CAFCASS) | Looks after the interests of children involved in family proceedings. Works under the rules of the Family Court and legislation to work with children and their families, then advise the courts on what is in the best interests of individual children. |
| Welsh Longitudinal General Practice Dataset (WLGP)—Welsh Primary Care | Diagnosis, medications, symptoms and procedures. |
| Welsh Demographic Service Dataset (WDSD) | People registered with a GP in Wales, gender, LSOA, deprivation score, Residential Linking Field, Date left Wales. |
| Geographic mapping | Environmental data including distance to facilities |
| Welsh Study of Mothers and Babies (WOMBS) | Ultrasound scan for cognitive development |
| COVID-19 vaccination and testing | COVID-19 vaccination uptake rates and tests |
| **Survey data** | |
| Born in Wales (BIW) surveys (Expectant Parent, 18–24 month, Nursery) | Well-being, stress, physical activity, ethnicity, nationality, relationship status, sexual orientation, mood/depression, alcohol consumption, smoking, weight/height, maternity care, breast feeding, local area services, occupation, education, income, brushing teeth, child activity and development. |
| Health and Attainment of Pupils in Primary EducatioN (HAPPEN) | Well-being, physical activity, diet, Me and My Feelings questionnaire (behavioural difficulty, emotional difficulty), garden, safety of area, hours of sleep, teeth brushing, ability to swim, ability to ride a bike. |

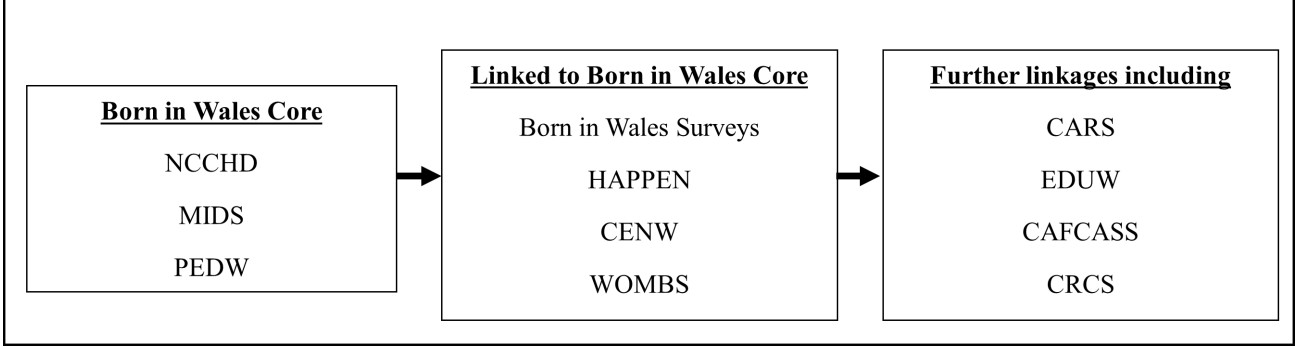

**Figure 1** Core components of Born in Wales and further linkages. CAFCASS, Children and Family Court Advisory and Support Service; CARS, Congenital Anomaly Register and Information Service; CENW, Census Wales; CRCS, Children Receiving Care and Support Census; EDUW, Education Wales; HAPPEN, Health & Attainment of Pupils in Primary EducatioN; MIDS, Maternal Indicators Dataset; NCCHD, National Community Child Health Database; PEDW, Patient Episode Dataset for Wales; WOMBS, Welsh Study of Mothers and Babies.

online. For both surveys, participants provide informed consent to be included in the cohort, and recruitment is ongoing. Over a 5-year period, we aim for an average of 1000 women/families per year to complete the surveys to further enrich the birth cohort. The data obtained from these surveys will be used to extrapolate (impute data) for the larger cohort study, enabling the estimation of variables such as income in the comprehensive all Wales cohort.

### Data-linkage hosting environment
The core datasets of Born in Wales consist of NCCHD, PEDW and MIDS. These datasets are enhanced by linking them to Census Wales, Welsh Study of Mothers and Babies and the survey data from Born in Wales and HAPPEN. Further linkages can include multiple datasets including Congenital Anomaly Register and Information Service, Education Wales, Children and Family Court Advisory and Support Service and Children Receiving Care and Support Census. The core components of Born in Wales and subsequent linkages are described in figure 1. The foundation of Born in Wales involves SAIL's robust anonymising system, which ensures secure data linkage in adherence to specified data protocols. As part of the data upload process to SAIL, the original dataset is divided into two distinct types of files. File 1 encompasses sensitive person-level demographics data, which is transmitted to Digital Health and Care Wales (DHCW). DHCW undertakes the processing, matching and anonymisation of the file 1 data before transmitting it to SAIL. File 2 comprises clinical data or other non-identifiable data, which is directly transmitted to SAIL (see flow diagram of this process in figure 2).

### Data-linkage procedures and resources
SAIL employs an Anonymous Linking Field (ALF) to establish connections with other records. Matching of two records arises when their respective ALFs are identical. Within the Born in Wales database, both maternal and child ALFs are used, enabling the identification and linkage of mothers to their corresponding children.

Consequently, this integration culminates in the creation of a comprehensive data repository, accessible to researchers who can request extracted data encompassing maternity, neonatal, and/or mental health services. The matching algorithm was collaboratively designed and assessed by a trustworthy third party and SAIL. The algorithm compares numerous personal identifiers between the received dataset and the Welsh Demographic Service Dataset (WDSD). The linkage is performed based on an National Health Service (NHS) number whenever possible (deterministic linkage). In the absence of an NHS number, a matching algorithm using surname, first name, post code, date of birth and sex is applied against the WDSD (probabilistic linkage). The algorithm's development ensures a high matching accuracy, with specific thresholds for match accuracy reported to SAIL within the anonymised dataset. A poor matching score indicates incomplete or inaccurate personal information provided by an individual or a lack of registration on the NHS database. The ALF system ensures a high level of match quality, as it facilitates the accurate matching of infants born within Born in Wales with their respective mothers' records.[18]

### Patient and public involvement
Patient and public involvement (PPI) is integral to the growth of Born in Wales, underscoring its commitment to inclusivity and participatory research. The inclusion of PPI ensures that members of the public actively participate in shaping the research project. Valuable input was sought during the design of the surveys, complemented by insights from midwives who shared their lived experiences on pertinent research inquiries. Moreover, a dedicated PPI steering group will be actively sought to incorporate feedback from the public. The PPI group will be recruited from midwives and health visitors via already established networks, along with expectant and new parents to help coproduce its delivery and approach from the survey to research questions and aims. Born in Wales aims to adopt a coproduction approach, emphasising collaborative partnerships among

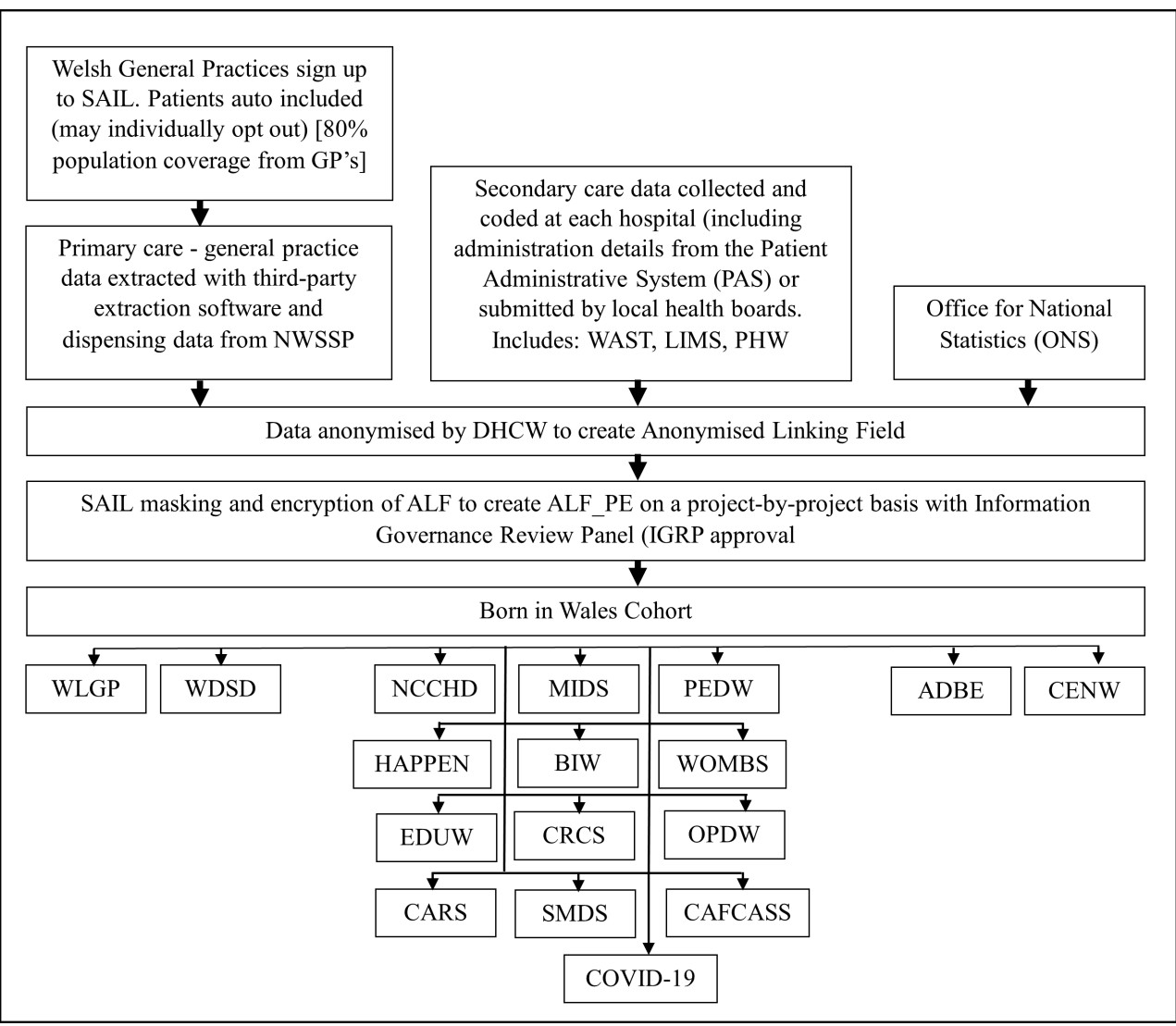

**Figure 2** Flow diagram of the SAIL process. ADBE, Annual District Birth Extract; ALF, Anonymised Linking Field; BIW, Born in Wales; CAFCASS, Children and Family Court Advisory and Support Service; CARS, Congenital Anomaly Register and Information Service; CENW, Census Wales; CRCS, Children Receiving Care and Support Census; DHCW, Digital Health and Care Wales; EDUW, Education Wales; HAPPEN, Health & Attainment of Pupils in Primary EducatioN; IGRP, Information Governance Review Panel; LIMS, Laboratory Information Management System; NCCHD, National Community Child Health Database; NHS, National Health Service; NWSSP, NHS Wales Shared Services Partnership; OPDW, Outpatient Database for Wales; PEDW, Patient Episode Dataset for Wales; PHW, Public Health Wales; SAIL, Secure Anonymised Information Linkage; SMDS, Substance Misuse Dataset; WAST, Welsh Ambulance Services NHS Trust; WDSD, Welsh Demographic Service Dataset; WLGP, Wales Longitudinal General Practice; WOMBS, Welsh Study of Mothers and Babies.

researchers, practitioners and the public. The decision-making processes within Born in Wales will prioritise PPI to ensure the inclusion of diverse perspectives. To facilitate this approach, Born in Wales will follow the Co-production of Research and Strategy standard operating procedure[21] and the UK standards for PPI involvement and National Institute for Health Research (NIHR) guidance from INVOLVE[22] which establishes a framework for meaningful engagement with practitioners and families. This approach fosters a reciprocal exchange, focusing on the important questions and concerns for stakeholders, while also enhancing the impact, translation and dissemination of findings. A record of PPI activity will be maintained using the Public Involvement in Research Impact Toolkit.[23]

## FINDINGS TO DATE
### Born in Wales cohort characteristics
The cohort comprises all children born in Wales since 2011 (from NCCHD) and followed until age 11. An electronic data spine has been established to encompass the entire population of children born in Wales in the last 13 years, resulting in a cohort size exceeding 400 000 children within Born in Wales. Furthermore, 2500 parents and 30 000 primary school children have been recruited to facilitate enriched data collection, all of whom are linked to the electronic cohort data spine. The Born in Wales electronic cohort encompasses the entirety of Wales, thus represents a national birth cohort for the country. In terms of gender distribution, females comprise 51%

 

of the cohort, while males account for 49%. 78.5% of the population in Wales is of Caucasian ethnicity with only 7.8% of the cohort from an ethnic minority background. Analysis of the NCCHD Births dataset reveals that 26.8% of children are below the age of 5, whereas 63.2% are aged between 5 and 11 years. Between 2011 to 2022, a total of n=435 031 births were recorded, involving n=269 421 mothers. Of these births, n=222 814 are male, n=212 198 are female births, and n=19 gender data is missing. Table 2 represents the characteristics of the electronic cohort.

| Table 2 | Characteristics of the cohort | | |
|---|---|---|---|
| **Characteristic** | **Value** | **N** | **Percentage (%)** |
| Mothers | Mothers | 269 421 | 100 |
| Babies | Babies | 435 031 | 100 |
| Sex (of baby) | Male | 222 814 | 51.22 |
| | Female | 212 198 | 48.78 |
| | Missing | 19 | 0.00 |
| Ethnic group (of mother) | White—any background | 211 728 | 78.49 |
| | Gypsy or Irish Traveller | 23 | 0.01 |
| | White and Black Caribbean | 883 | 0.33 |
| | White and Black African | 1153 | 0.43 |
| | White and Asian | 1153 | 0.43 |
| | Any other mixed background/multiple ethnic background | 3209 | 1.19 |
| | Indian | 1922 | 0.71 |
| | Pakistani | 1750 | 0.65 |
| | Bangladeshi | 1439 | 0.53 |
| | Chinese | 2284 | 0.85 |
| | Any other Asian background | 262 | 0.10 |
| | Caribbean | 2156 | 0.80 |
| | African | 568 | 0.21 |
| | Any other black background | 793 | 0.29 |
| | Any other ethnic group | 3396 | 1.26 |
| | Arab | 14 | 0.01 |
| | Missing | 37 026 | 13.73 |
| Stillbirths | Live birth | 433 315 | 99.61 |
| | Stillbirth | 1716 | 0.39 |
| Gestation age category | Extremely premature (EPT) | 2278 | 0.52 |
| | Very premature (VPT) | 3552 | 0.82 |
| | Premature (PT) | 25 809 | 5.93 |
| | Term (T) | 345 277 | 79.37 |
| | Late term (LT) | 15 738 | 3.62 |
| | Very late term (VLT) | 285 | 0.07 |
| | Missing | 42 092 | 9.68 |
| Birth weight category | Extremely low birth weight (ELBW) | 35 333 | 8.12 |
| | Very low birth weight (VLBW) | 2249 | 0.52 |
| | Low birth weight (LBW) | 21 179 | 4.87 |
| | Normal birth weight (NBW) | 296 585 | 68.18 |
| | High birth weight (HBW) | 36 877 | 8.48 |
| | Very high birth weight (VHBW) | 5897 | 1.36 |
| | Missing | 36 911 | 8.48 |
| Singleton or multiple birth | Multiple | 11 656 | 2.68 |
| | Singleton | 408 456 | 93.89 |
| | Missing | 14 919 | 3.43 |

## Maternal, birth and birth outcomes

Between 2011 to 2022, there were n=1716 stillbirths, constituting 0.39% of all births. Examining gestational age, 7.27% (n=31 639) of infants were born preterm, defined as delivery before 37 weeks gestation, while 83.06% (n=3 61 300) were born at term or later (37 weeks and onwards).

The Born in Wales database has recently been used to investigate the risk factors associated with low birth weight (LBW), defined as a birth weight of less than 2500 g. Analyses revealed that non-singleton children exhibited the highest risk of LBW (OR 21.74 (95% CI 21.09 to 22.40)), followed by pregnancy intervals of less than 1 year (OR 2.92 (95% CI 2.70 to 3.15)).[24] Maternal physical and mental health conditions, including diabetes (OR 2.03 (95% CI 1.81 to 2.28)), anaemia (OR 1.26 (95% CI 1.16 to 1.36)), depression (OR 1.58 (95% CI 1.43 to 1.75)), serious mental illness (OR 1.46 (95% CI 1.04 to 2.05)), anxiety (OR 1.22 (95% CI 1.08 to 1.38)) and use of antidepressant medication during pregnancy (OR 1.92 (95% CI 1.20 to 3.07)) were also identified as significant risk factors for LBW. Additional maternal risk factors include smoking (OR 1.80 (95% CI 1.76 to 1.84)), alcohol-related hospital admission (OR 1.60 (95% CI 1.30 to 1.97)), substance misuse (OR 1.35 (95% CI 1.29 to 1.41)) and evidence of domestic abuse (OR 1.98 (95% CI 1.39 to 2.81)).[24] Numbers and percentages are provided in the online supplemental information.

## Born in Wales survey participants

Two thousand five hundred parents have completed the Born in Wales surveys. The mean age of Born in Wales survey participants was 32 years (SD=12.5), with an IQR of 28–35 years. The Born in Wales survey data were used in a mixed methods study conducted during the COVID-19 pandemic, aiming to examine the effects of the pandemic on pregnancy experiences and birth outcomes in 2020. Results indicated that the pandemic had a notable adverse effect on the psychological well-being of 71% of survey respondents. These individuals reported heightened levels of anxiety, stress and feelings of loneliness, which were associated with attending prenatal scans in the absence of their partner, giving birth alone, and restricted interactions with midwives. However, no significant differences were observed in annual birth outcomes, including gestation, birth weight, stillbirths and Caesarean sections, between infants born in 2020 compared with 2016–2019. While the pandemic negatively affected mothers' pregnancy experiences, population-level data indicates that this did not result in adverse birth outcomes for infants born during the pandemic.[25]

## HAPPEN children's primary school network

The HAPPEN network represents a substantial cohort of over 30 000 school-aged children across Wales, with a gender distribution of approximately 47% boys, 49% girls, and 3% who preferred not to disclose their gender. The average age of the children is 9.35 years. Recently, the HAPPEN dataset has been used to explore the health and well-being of children during the COVID-19 pandemic. This encompassed a retrospective cohort study employing an online cohort survey conducted between January 2018 to February 2020, in conjunction with routine PCR SARS-CoV-2 test results. The study explored health-related behaviours in children spanning the period from 2018 to 2020, and their association with being tested and testing positive in 2020 to 2021. Notably, the investigation revealed significant associations between parental health literacy and monitoring behaviours.[26] Furthermore, the study employed free school meal status (FSM) as a proxy for deprivation in an exploratory analysis examining the impact of school closures on the health and well-being. The findings indicated that children eligible for FSM may experience adverse consequences in terms of physical health, including reduced physical activity and suboptimal dietary choices, as a result of prolonged school closures.[27]

## DISCUSSION

### Strengths and limitations

Born in Wales has established a comprehensive, Wales-wide population-based database which consolidates clinical data from maternity, neonatal, child health and education records. This national-scale database is supplemented by quantitative and qualitative results from surveys conducted by Born in Wales, providing rich insights into details that cannot be obtained through routinely-collected data. The existence of this database enables further data linkage, facilitating life course research on the health and well-being of the Wales population. This has significant implications for enhancing healthcare delivery at the local level and offering valuable research evidence to guide policy and practise both within the studied population and in other countries. This research aims to generate robust evidence that can influence policy and practice, enabling the implementation of early interventions to promote child health and well-being. Moreover, collaborations with diverse cohorts facilitate comparisons across different populations, enabling a broader understanding of factors influencing child health and well-being outcomes.

One inherent constraint associated with routine and administrative data is the presence of missing data or errors. However, SAIL has established a robust framework that includes a team of skilled analysts and rigorous quality control procedures to address issues such as duplication of patient data entries and minimise instances of missing data. Subsequently, the Born in Wales project can contribute to the enhancement of clinical reporting practice and bolster the reliability of research findings derived from the database. Additionally, the utilisation of anonymised cohorts serves as an effective strategy for overcoming the barriers related to obtaining consent from individuals, thereby facilitating the seamless aggregation and analysis of data.

A potential constraint of Born in Wales is the loss of data pertaining to individuals who relocate outside of Wales during pregnancy or after the child's birth. Analogous to other datasets, research conducted utilising Born in Wales may be constrained by the information available through routine data collection. Nevertheless, Born in Wales can enhance the available information by incorporating survey questions that capture data missing from administrative records. It is important to acknowledge that data entry can be subject to human error, and there may still be instances of missing records.

### Research aspirations

As the Born in Wales database continues to expand, there are intentions to broaden the scope by incorporating additional health, social care, and education data, alongside Police data. The subsequent phase of data linkage will involve integrating Police data into SAIL. Replicating this data linkage model on a national scale within the UK and potentially extending it globally would enable the establishment of more extensive research cohorts and facilitate cross-comparisons across diverse populations.

This cohort is established through the integration of routine data records, encompassing health and administrative data, as well as environmental data supplemented with survey data to enrich the cohort. It constitutes a comprehensive, data rich total population cohort constructed through substantial investments and extensive record linkage efforts from HDR UK and ADR. Furthermore, this methodology demonstrates cost-effectiveness by leveraging existing data sources, establishing a dynamic cohort that continuously expands its participant pool. The cohort is well suited for investigating natural experiments and interventions. The endeavours undertaken in this study closely align with recommendations 1–3 and 6 of the MRC strategy to maximise UK population cohorts.[28]

The design of this cohort facilitates cross-cohort comparisons with other electronic cohorts in the UK, including BiB[3] and eLIXIR.[4] To harmonise phenotypic variables across studies and adopt core common data standards, all data in the cohort will adhere to the Observational Medical Outcomes Partnership standard. Knowledge sharing and directories have been facilitated through the HDR UK gateway, and high-quality, robust meta-data is provided, adhering to recommendations 4 and 5 of the MRC strategy.[28]

This cohort is unique due to the comprehensive breadth of routine and survey data collected and linked in a total population cohort. It is designed to enable data sharing by employing harmonised data and meta-data, following the principles of findability, accessibility, interoperability, and reusability. A distinguishing feature of Born in Wales is its incorporation of census data linked to numerous datasets, encompassing social care and justice datasets. Additionally, the database includes self-reported data which captures aspects that are not typically electronically captured, such as stress, well-being, physical activity, access to services and use of services.

### Data availability and access

The data for the Born in Wales cohort is available in the SAIL Databank at Swansea University, Swansea, UK.[29] All proposals to use SAIL data are subject to review by an independent Information Governance Review Panel (IGRP). This project's approval code is 0916. Before any data can be accessed, approval must be given by the IGRP. To use Born in Wales data, you need to provide a safe researcher training certificate, a signed data access agreement and IGRP approval.

### Future directions

A comprehensive research database has been established, encompassing data pertaining to maternity, neonatal and child health. This database integrates information on maternal and paternal health throughout pregnancy and extends to encompass the subsequent health of the child. Moreover, it enriches the dataset with valuable qualitative insights obtained from the Born in Wales surveys, which capture nuanced information not captured solely by hospital records. Born in Wales is committed to continuous progress and expansion through collaborative efforts, enabling longitudinal follow-up of families in Wales and fostering coproduction of research questions relevant to each stage of development. By encompassing the entire population of Wales, Born in Wales generates a wealth of rich data on the national scale, which has the potential to inform interventions aimed at promoting healthy growth and development among the population.

**Correction notice** This article has been corrected since it was published. Licence updated to CC BY on 2nd August 2024.

**Contributors** All authors provided substantial contributions to the piece of work. HJ, SB and SDr contributed to the conception of this article. MJS, NLK, MJ, AmB, AdB and SDa were involved in manuscript writing and revision. HJ, MJS, AmB and MJ were involved in data analysis and interpretation. HJ is responsible for the overall content as the guarantor. All authors read and approved the final manuscript. All authors agree to be accountable for all aspects of the work.

**Funding** This work was supported by the National Centre for Population Health and Wellbeing Research (NCPHWR) grant number [AMS103836].

**Competing interests** None declared.

**Patient and public involvement** Patients and/or the public were involved in the design, or conduct, or reporting, or dissemination plans of this research. Refer to the Cohort description section for further details.

**Patient consent for publication** Not applicable.

**Ethics approval** This study involves human participants and was approved by HRA Approval June 2021 REC Reference 21/NW/0156IRAS 299760. Participants gave informed consent to participate in the study before taking part.

**Provenance and peer review** Not commissioned; externally peer reviewed.

**Data availability statement** Data are available upon reasonable request. Data are available upon reasonable request. Researchers can apply for data access by submitting a research application to the SAIL team. The SAIL website provides information on the application process (https://saildatabank.com/data/apply-to-work-with-the-data/). All proposals to use SAIL data are subject to review by an independent Information Governance Review Panel (IGRP). This project's approval code is 0916. Before any data can be accessed, approval must be given by the

IGRP. To use Born in Wales data you need to provide a safe researcher training certificate, a signed data access agreement and IGRP approval.

**ORCID iDs**
Hope Jones http://orcid.org/0000-0003-4312-476X
Mike J Seaborne http://orcid.org/0000-0002-4921-7556
Michaela James http://orcid.org/0000-0001-7047-0049
Amrita Bandyopadhyay http://orcid.org/0000-0003-2798-4030
Adele Battaglia http://orcid.org/0000-0002-6737-0343
Sinead Brophy http://orcid.org/0000-0001-7417-2858

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
