## [Reviewer comments · BMJ Open]

ARTICLE DETAILS

TITLE (PROVISIONAL)	Cohort Profile: Born in Wales - a birth cohort with maternity, parental, and child data linkage for life course research in Wales, UK
AUTHORS	Jones, Hope; Seaborne, Michael; Kennedy, Natasha; James, M; Dredge, Sam; Bandyopadhyay, Amrita; Battaglia, Adele; Davies, Sarah; Brophy, Sinead

VERSION 1 – REVIEW

REVIEWER	Lee, Seung Won Sejong University, Department of Data Science
REVIEW RETURNED	16-Sep-2023

GENERAL COMMENTS	Dear Authors, I have completed my review of your manuscript, titled "Cohort Profile: Born in Wales - A Birth Cohort with Maternity, Parental, and Child Data Linkage for Life Course Research in Wales, UK." The study provides valuable descriptive information on a Wales birth cohort, collected from 2011 to 2023, utilizing Secure Anonymised Information Linkage (SAIL). Overall, the manuscript is well-conceived and generally well-written. However, I have some minor suggestions that could enhance the quality of the manuscript: 1. Inconsistency in Data Description: The manuscript mentions that descriptive data were collected from 2011 to 2023 but does not clarify whether this includes health, education, and social care data consistently across all those years. Clarity on this aspect would help the reader understand the completeness of the dataset.2. Participants Section: The manuscript indicates that 2,500 parents and 30,000 primary school children have been recruited for "enhanced data collection." Could you please clarify what is meant by "enhanced" in this context? It would also be helpful to know how these participants differ from the 400,000 child electronic records previously mentioned.3. Ethnicity Data: While you note that 6% of the cohort comes from ethnic minority backgrounds, the specific ethnic groups are not detailed. Providing this information could offer greater insight into the cohort's diversity and its broader applicability.
--

	4. Methodological Details: The term "data linkage techniques" is somewhat vague. More detailed explanation about the methods used to ensure data quality and reliability would strengthen this section. Additionally, it would be valuable to elaborate on the robust data security and governance frameworks mentioned. 5. Future Plans: While the manuscript does discuss future research aims and expansion plans, it would be beneficial to specify how the dataset may be shared with other researchers, and whether there are plans for external validation. Furthermore, if there are preliminary findings or trends, including these would add depth to the manuscript. In summary, I would like to recommend the manuscript for acceptance. Thank you for authors' valuable contribution to our scientific community. I look forward to reviewing the revised manuscript.
--	--

REVIEWER	Araki, Atsuko Hokkaido Daigaku, Faculty of Health Sciences
REVIEW RETURNED	22-Oct-2023

GENERAL COMMENTS	This is a manuscript of birth cohort "Born in Wales" profile paper. In the manuscript, the authors provided information about the sets of data linked for Born in Wales. The authors also provide basic descriptive information of "Born in Wales". The reviewer agree the strength of "Born in Wales" as a cohort with different data linkages. Consolidate of clinical and national data from maternity, neonatal, child health and education and all potential linkage for the lifetime records will provide broad information for different research questions. On the other hand, the manuscript is somewhat difficult to follow. The authors consolidate different dataset, but it is not clear how the reported number collected from which dataset. It is not clear about the connection between each number given from which dataset. The reviewer suggests reconstructing the manuscript for clarification. In addition, please provide the ethical statement in the manuscript. It is not clear what is the main purpose of this manuscript. As a data sources, the authors listed all potential data to be consolidate. The basic data (sample size over 400000) is WLGP referring all annual birth as the electronic birth cohort? In the abstract, "Participants" section, it is mentioned 400,000 data for the Born in Wales". However, the "findings to date" is about 2500 parents and 30000 primary school children's distribution. Please match the information given in the "participants" and "findings to date". P8 it is mentioned "over a 5 year period, an average of 1000 women/families per year will be targeted for recruitment to further enrich the birth cohort". The structure of this participants with other data set is not clear. Are these 1000 from 30,000 birth every year? How are they selected? If this part is not initiated yet, when the authors plan to start?
--

	P13 “The PPI group will be recruited from midwives and health visitors.” How they will be selected? Is this different from the above mentioned 100 women/families per year? P14 “Born in Wales cohort characteristics.” Please clarify when recruiting has been initiated and if the inclusion still ongoing or terminated for each explanation. The reviewer suggests to include one table with the participants characteristics and birth outcomes of “Born in Wales so far, written from line 7 to line 19. Line 4 to 5, “2500 parents and 30000 primary school children have been recruited to facilitate enhanced data collection” How they are recruited, when the recruitment start? Are they provided informed consent to be included in the cohort, or selected from the dataset for linkage? Please clarify the name of the data mentioned in the Table 1 for these two participants. L10 “Between 2011 to 2022” Does this mean from January 2011 to December 2022? Please clarify. Is the number of N=435032 about 80% of total birth after excluding those who refuse to use the data via optout? Please clarify. Maternal, birth and birth outcomes Is the total sample of this information is 435032 birth? Please clarify. Please also provide the number and % for late delivery (37 weeks and onward). Please clarify LBW is <2500g. Please provide non-singleton children’s number and %. Are there any miscarriage and still birth included in the dataset or only live birth? Please include such information. The reviewer suggests providing all number % of relevant information related to odds ratio given in the table, e.g. pregnancy intervals, diabetes, anemia, depression, serious mental illness, anxiety, antidepressant medication, smoking, alcohol-related hospital admission, substance misuse and domestic abuse. In addition, is this information collected all from database, or from “born in Wales survey participants”? Born in Wales survey participants. Who are the participants? Recruited as PPI as additional 1000 per year? Do all information including “anxiety, stress, abuse, etc. collected by questionnaire base? Did the authors use specific existing questionnaires? Please clarify what “negative affected mothers’ pregnancy experiences”. There is no ethical statement included in this manuscript. In addition, did the authors collect all informed consent through optout? Please clarify. Minor comments P8 last sentence “It is important to note that 94% of the population in Wales is of Caucasian ethnicity.” The editor think the sentence not fit well here. Please consider moving this sentence to the section “born in Wales cohort characteristics”, close to the part “6% of the cohort are from an ethnic minority background.
--	---

	P19 L5 Please spell out "OMOP". End.
--	--

VERSION 1 – AUTHOR RESPONSE

Reviewer: 1

I have completed my review of your manuscript, titled "Cohort Profile: Born in Wales - A Birth Cohort with Maternity, Parental, and Child Data Linkage for Life Course Research in Wales, UK." The study provides valuable descriptive information on a Wales birth cohort, collected from 2011 to 2023, utilizing Secure Anonymised Information Linkage (SAIL). Overall, the manuscript is well-conceived and generally well-written. However, I have some minor suggestions that could enhance the quality of the manuscript:

1. Inconsistency in Data Description: The manuscript mentions that descriptive data were collected from 2011 to 2023 but does not clarify whether this includes health, education, and social care data consistently across all those years. Clarity on this aspect would help the reader understand the completeness of the dataset.

Thank you for this comment. Amendments have been made to the manuscript to clarify where the descriptive data comes from. The following has been added: 'Descriptive information from 2011 to 2023 has been gathered from the National Community Child Health Database (NCCHD) in SAIL.' (Page 2)

Further linkages permit a wide range of health, social, and education analyses to take place. While there is some inconsistency with temporal coverage across all linkable datasets and inevitable differences in availability of data for each individual, it is not the intention of Born in Wales to create a cohort which is holistic and complete; only to provide a core dataset which can later be linked to and used as a resource for future research. With respect to education, the temporal coverage will match that of the Born in Wales cohort, however some of the care data (e.g. Children Receiving Care and Support (CRCS)) only commenced in 2016. These would need to be considered when designing a research project around such data.

2. Participants Section: The manuscript indicates that 2,500 parents and 30,000 primary school children have been recruited for "enhanced data collection." Could you please clarify what is meant by "enhanced" in this context? It would also be helpful to know how these participants differ from the 400,000 child electronic records previously mentioned.

Thank you for this comment. The 2,500 parents and 30,000 primary school children have all completed surveys which provide information that is not collected in the routine data (stress, wellbeing etc). This is what is meant by 'enhanced' as this data is not captured in the routinely collected data. This terminology has been amended to 'enriched' as the surveys aim to add valuable personal insights. These survey participants will be present in the population-level data (400,000) and represent a cross-section of the Wales population.

3. Ethnicity Data: While you note that 6% of the cohort comes from ethnic minority backgrounds, the specific ethnic groups are not detailed. Providing this information could offer greater insight into the cohort's diversity and its broader applicability.

Details of the ethnic groups have been added in Table 2 (Page 16). It must be noted that the percentage and numbers vary from the first manuscript based on more successful identification of ethnic groups from various datasets.

4. Methodological Details: The term "data linkage techniques" is somewhat vague. More detailed explanation about the methods used to ensure data quality and reliability would strengthen this section. Additionally, it would be valuable to elaborate on the robust data security and governance frameworks mentioned.

This has been amended to clarify what data linkage techniques were used. The following has been added

'Data of anonymised identified persons were linked on the individual level to data from health, social care and education data within the Secure Anonymised Information Linkage (SAIL) databank. Each individual is assigned an encrypted anonymised linking field; this field is used to link anonymised individuals across datasets.' (Page 2)

The robust data security and governance frameworks are elaborated on later in the manuscript under the heading Data-linkage procedures and resources (see below).

'The matching algorithm was collaboratively designed and assessed by a trustworthy third party and SAIL. The algorithm compares numerous personal identifiers between the received dataset and the Welsh Demographic Service Dataset (WDS). The linkage is performed based on an NHS number whenever possible (deterministic linkage). In the absence of an NHS number, a matching algorithm utilising surname, first name, post code, date of birth, and sex is applied against the WDS (probabilistic linkage). The algorithm's development ensures a high matching accuracy, with specific thresholds for match accuracy reported to SAIL within the anonymised dataset. A poor matching score indicates incomplete or inaccurate personal information provided by an individual or a lack of registration on the NHS database. The ALF system ensures a high level of match quality, as it facilitates the accurate matching of infants born within Born in Wales with their respective mothers' records [18].' (Page 12)

5. Future Plans: While the manuscript does discuss future research aims and expansion plans, it would be beneficial to specify how the dataset may be shared with other researchers, and whether there are plans for external validation. Furthermore, if there are preliminary findings or trends, including these would add depth to the manuscript.

Thank you for this comment. The data availability statement explains how other researchers can access the dataset (see below).

'Data availability statement: Data are available upon reasonable request. Researchers can apply for data access by submitting a research application to the SAIL team. The SAIL website provides information on the application process (<https://saildatabank.com/data/apply-to-work-with-the-data/>). All proposals to use SAIL data are subject to review by an independent Information Governance Review Panel (IGRP). This project's approval code is 0916. Before any data can be accessed, approval must be given by the IGRP. To use Born in Wales data you need to provide a safe researcher training certificate, a signed data access agreement and IGRP approval.' (Page 24)

A list of all the datasets are available on the Health Data Research (HDR) gateway (<https://www.healthdatagateway.org/>). The metadata is also available on the HDR gateway (<https://web.www.healthdatagateway.org/dataset/e7058fce-7c4f-4969-8850-172025bf851c>). External validation would be via birth statistics published by ONS. We are using what the Welsh government uses to derive their statistics <https://www.gov.wales/maternity-and-birth-statistics-2020.html>, so we can compare with their reports.

As for preliminary trends, this document aims solely to present a summary of the breadth of data available and scope for this birth cohort; findings will emerge in subsequent research projects using this data.

In summary, I would like to recommend the manuscript for acceptance. Thank you for authors' valuable contribution to our scientific community. I look forward to reviewing the revised manuscript.

Reviewer: 2

This is a manuscript of birth cohort "Born in Wales" profile paper. In the manuscript, the authors provided information about the sets of data linked for Born in Wales. The authors also provide basic descriptive information of "Born in Wales".

The reviewer agree the strength of "Born in Wales" as a cohort with different data linkages. Consolidate of clinical and national data from maternity, neonatal, child health and education and all potential linkage for the lifetime records will provide broad information for different research questions.

On the other hand, the manuscript is somewhat difficult to follow. The authors consolidate different dataset, but it is not clear how the reported number collected from which dataset. It is not clear about the connection between each number given from which dataset. The reviewer suggests reconstructing the manuscript for clarification.

Thank you for this comment. Amendments have been made to clarify this (see below).

'Descriptive information from 2011 to 2023 has been gathered from the National Community Child Health Database (NCCHD) in SAIL.' (Page 2). For clarity, the Born in Wales dataset has been constructed using all data available from NCCHD. This has been linked to other related datasets for the purpose of completing missing data, to identify and correct erroneous data, and to provide additional maternity/birth data which is not necessarily recorded in NCCHD. While there is additional information provided in this document about other linkable datasets, these have not been used to create this core dataset but are indicative of their availability for future research. A full breakdown of which dataset has contributed to this cohort construction is not currently available. In essence, the data within NCCHD forms the spine of the Born in Wales cohort, while other linked datasets fill missing data and form the ribs of the cohort. As such, the overall number of individuals within the Born in Wales cohort matches the number within NCCHD.

In addition, please provide the ethical statement in the manuscript.

The following ethical statement has been added

'This study was approved by the SAIL Databank independent Information Governance Review Panel (IGRP project number 0916, Wales Electronic Cohort for Children Phase 4). The survey elements were approved by the Health Research Authority (protocol number RIO 030-20).' (Page 14)

It is not clear what is the main purpose of this manuscript. As a data sources, the authors listed all potential data to be consolidate. The basic data (sample size over 400000) is WLGP referring all annual birth as the electronic birth cohort?

The following has been added to the introduction to indicate the purpose of the manuscript.

'This document outlines how we have used the NCCHD as a core dataset to build a more comprehensive and complete maternity and birth cohort in Wales for future longitudinal life-course research analyses.' (Page 7)

The core data (400,000) has been clarified as coming from the NCCHD (Page 2).

In the abstract, "Participants" section, it is mentioned 400,000 data for the Born in Wales". However, the "findings to date" is about 2500 parents and 30000 primary school children's distribution. Please match the information given in the "participants" and "findings to date".

Thank you for this comment. To clarify, the 400,000 represents all births in Wales from 2011, providing a national database, while the survey responses represent a cross section of the population. Edits have been made for clarity (see below)

The participants section now states 'survey responses about health and wellbeing from a cross-section of the population including 2,500 parents and 30,000 primary school children have been collected for enriched personal responses and linkage to the data spine.' (Page 2)

The findings to date section now states 'The cohort comprises all children born in Wales since 2011, with follow-up conducted until they finish primary school at age 11. The child cohort is 51%: 49% female: male, and 7.8% are from ethnic minority backgrounds. When considering age distribution, 26.8% of children are under the age of 5, while 63.2% fall within the age range of 5-11.' (Page 2)

P8 it is mentioned "over a 5 year period, an average of 1000 women/families per year will be targeted for recruitment to further enrich the birth cohort". The structure of this participants with other data set is not clear. Are these 1000 from 30,000 birth every year? How are they selected? If this part is not initiated yet, when the authors plan to start?

Amendments have been made to clarify this (See below)

'Over a 5 year period, we aim for an average of 1,000 women/families per year to complete the surveys to further enrich the birth cohort.' (Page 9)

To clarify, we aim to recruit 1,000 parents each year to complete surveys. Survey participants are recruited via social media adverts including Facebook and Instagram. These adverts run continuously so we are recruiting continually. These 1000 respondents are parents so they will not be part of the 30,000 new births but they may be parents of those new babies.

P13 "The PPI group will be recruited from midwives and health visitors." How they will be selected? Is this different from the above mentioned 100 women/families per year?

The manuscript states 'The PPI group will be recruited from midwives and health visitors via already established networks, along with expectant and new parents to help co-produce its delivery and approach from the survey to research questions and aims.' (Page 13). We have established relationships with local midwives and research midwives across Wales who will be invited to be part of the PPI group. Parents will be invited to express their interest in being a PPI member and join the group.

To clarify, the PPI group will be different from the 1,000 families per year participating in the surveys. PPI members ranging from those who work in women's health, child health, child education and members of the public will inform future research questions, how we disseminate our results etc.

P14 "Born in Wales cohort characteristics."

Please clarify when recruiting has been initiated and if the inclusion still ongoing or terminated for each explanation. The reviewer suggests to include one table with the participants characteristics and birth outcomes of "Born in Wales so far, written from line 7 to line 19.

The following detail has been added to clarify this.

'Since 2016, 30,000 primary school children have completed the Health & Attainment of Pupils in Primary Education (HAPPEN) Survey. Primary schools are recruited via e-mail and local authority referral. The survey is completed online, in school, at a time that suits the school. The survey has been developed alongside the new school curriculum for Wales priorities for health and wellbeing. Since 2020, 2,500 parents have been recruited via social media to complete the Born in Wales surveys. This survey is also completed online. For both surveys, participants provide informed consent to be included in the cohort, and recruitment is ongoing.' (Page 9)

The cohort characteristics have been added in Table 2 (Page 15-17).

Line 4 to 5, “2500 parents and 30000 primary school children have been recruited to facilitate enhanced data collection” How they are recruited, when the recruitment start? Are they provided informed consent to be included in the cohort, or selected from the dataset for linkage? Please clarify the name of the data mentioned in the Table 1 for these two participants.

The following has been added to the manuscript for clarity.

‘Since 2016, 30,000 primary school children have completed the Health & Attainment of Pupils in Primary Education (HAPPEN) Survey. Primary schools are recruited via e-mail and local authority referral. The survey is completed online, in school, at a time that suits the school. The survey has been developed alongside the new school curriculum for Wales priorities for health and wellbeing. Since 2020, 2,500 parents have been recruited via social media to complete the Born in Wales surveys. This survey is also completed online. For both surveys, participants provide informed consent to be included in the cohort, and recruitment is ongoing.’ (Page 9).

Subheadings in Table 1 (Page 9-11) have been added to identify the electronic data and the survey data. The survey participants are from ‘Born in Wales (BIW) surveys (Expectant Parent, 18–24-month, Nursery)’ and ‘Health & Attainment of Pupils in Primary Education (HAPPEN)’

L10 “Between 2011 to 2022” Does this mean from January 2011 to December 2022? Please clarify.

Yes that is correct, these dates have been included for clarity.

Is the number of N=435032 about 80% of total birth after excluding those who refuse to use the data via optout? Please clarify.

The dataset defined in this paper is built upon the NCCHD as its core and thus the N=435,031 is the number of registered births in Wales between the specified dates. (one less birth was identified when we re-ran the code for this revision). This is the population of births in its entirety as there is no opt-out for this core dataset.

Maternal, birth and birth outcomes

Is the total sample of this information is 435032 birth? Please clarify.

The total sample of this information is N=435,031 (one less birth was identified when we re-ran the code for this revision).

Please also provide the number and % for late delivery (37 weeks and onward).

The number and % for all gestational age categories have been provided in Table 2 (Page 16-17).

Please clarify LBW is <2500g.

This has been amended (see below).

‘The Born in Wales database has recently been utilised to investigate the risk factors associated with low birth weight (LBW), defined as a birth weight of less than 2500g’ (Page 18)

Please provide non-singleton children’s number and %.

The number and % of non singleton and singleton births have been added in Table 2 (Page 17).

Are there any miscarriage and still birth included in the dataset or only live birth? Please include such information.

Still births are included in the dataset. The manuscript states 'Between 2011 to 2022, there were N=1,716 stillbirths, constituting 0.39% of all births' (Page 17). However, miscarriage is not included in the dataset.

The reviewer suggests providing all number % of relevant information related to odds ratio given in the table, e.g. pregnancy intervals, diabetes, anemia, depression, serious mental illness, anxiety, antidepressant medication, smoking, alcohol-related hospital admission, substance misuse and domestic abuse.

Thank you for this comment. A table providing the number and % of this information has been added to the supplementary information.

In addition, is this information collected all from database, or from "born in Wales survey participants"? To clarify, this information is from the population level NCCHD data not from the survey responses.

Born in Wales survey participants.

Who are the participants? Recruited as PPI as additional 1000 per year? Do all information including "anxiety, stress, abuse, etc. collected by questionnaire base? Did the authors use specific existing questionnaires?

The Born in Wales survey participants are the 2,500 parents recruited via social media adverts. These participants do not form part of the PPI group; they are solely survey respondents. The survey questions ask about health and wellbeing including anxiety and stress during pregnancy. The surveys are not previously existing questionnaires.

Please clarify what "negative affected mothers' pregnancy experiences".

To clarify, qualitative survey responses from expectant mothers showed that they were anxious, stressed and lonely during their pregnancy due to the covid-19 restrictions. These findings showed that the restrictions had a negative impact on the mental health of mothers during their pregnancy.

There is no ethical statement included in this manuscript.

The following ethical statement has been added.

'This study was approved by the SAIL Databank independent Information Governance Review Panel (IGRP project number 0916, Wales Electronic Cohort for Children Phase 4). The survey elements were approved by the Health Research Authority (protocol number RIO 030-20).' (Page 14)

In addition, did the authors collect all informed consent through optout? Please clarify.

The data used in this study are available in the SAIL databank [29] at Swansea University. All data held in the SAIL databank are anonymised; therefore, ethical approval is not mandatory in accordance with the Health Research Authority guidance and there is no legal requirement for explicit consent to participate under the Data Protection Act and UK GDPR. Additionally, individuals in a participating practice can request for their data to be removed from SAIL, this is undertaken by a trusted third party due to the anonymous nature of SAIL. Furthermore, permission has been obtained from the relevant Caldicott Guardian or Data Protection Officer for all data contained in SAIL. In addition, proposals using SAIL data are subject to review by an Information Governance Review Panel (IGRP) to secure approval. The IGRP approval number for this study is 0916.

Minor comments

P8 last sentence “It is important to note that 94% of the population in Wales is of Caucasian ethnicity.”
The editor think the sentence not fit well here. Please consider moving this sentence to the section
“born in Wales cohort characteristics”, close to the part “6% of the cohort are from an ethnic minority
background.

Thank you for this comment. This sentence has been moved as recommended.

P19 L5 Please spell out “OMOP”.

This has been amended to state ‘Observational Medical Outcomes Partnership (OMOP)’ (Page 22)

VERSION 2 – REVIEW

REVIEWER	Araki, Atsuko Hokkaido Daigaku, Faculty of Health Sciences
REVIEW RETURNED	28-Dec-2023
GENERAL COMMENTS	The authors modified the manuscript accordingly to clarify the reviewer's comments. I do not have any further comments.

VERSION 2 – AUTHOR RESPONSE